# Experimental Investigation of the Novel Periodic Feed Pressure Technique in Minimizing Fouling during the Filtration of Oily Water Systems Using Ceramic Membranes

**DOI:** 10.3390/membranes12090868

**Published:** 2022-09-08

**Authors:** Mohamed Echakouri, Amgad Salama, Amr Henni

**Affiliations:** Process System Engineering, Produced Water Treatment Laboratory, Faculty of Engineering and Applied Science, University of Regina, Regina, SK S4S 0A2, Canada

**Keywords:** oily wastewater, ceramic membrane, fouling mitigation, fouling resistance, periodic feed pressure technique

## Abstract

Fouling represents a bottleneck problem for promoting the use of membranes in filtration and separation applications. It becomes even more persistent when it comes to the filtration of fluid emulsions. In this case, a gel-like layer that combines droplets, impurities, salts, and other materials form at the membrane’s surface, blocking its pores. It is, therefore, a privilege to combat fouling by minimizing the accumulation of these droplets that work as seeds for other incoming droplets to cluster and coalesce with. In this work, we explore the use of the newly developed and novel periodic feed pressure technique (PFPT) in combating the fouling of ceramic membranes upon the filtration of oily water systems. The PFPT is based on alternating the applied transmembrane pressure (TMP) between the operating one and zero. A PFPT cycle is composed of a filtration half-cycle and a cleaning half-cycle. Permeation occurs when the TMP is set at its working value, while the cleaning occurs when it is zero. Three PFPT patterns were examined over two feeds of oily water systems with oil contents of 100 and 200 ppm, respectively. The results show that the PFPT is very effective in minimizing the problem of fouling compared to a non-PFPT normal filtration. Furthermore, the overall drops in permeate flux during the cleaning half-cycles are compensated by appreciable enhancement due to the significant elimination of fouling development such that the overall production of filtered water is even increased. Inspection of the internal surface of the membrane post rinsing at the end of the experiment proves that all PFPT cycles maintained the ceramic membranes as clean after a 2-h operation. This can ensure a prolonged lifespan of the ceramic membrane use and a continuous greater permeate volume production. The advantage of the PFPT is that it can be implemented on existing units with minimal modification, ease of operation, and saving energy.

## 1. Introduction

Produced water is a class of wastewater produced alongside several industries, including crude oil and natural gas, textile, food processing, pharmaceuticals, and others. Produced water is a complex mixture that contains suspended solids, heavy metals, dissolved oil and gasses, bacteria, insoluble oil, organics, brine, and several others [1]. Due to its polluting nature and the large volume produced, it requires effective treatment technologies. Filtration of produced water has recently received increasing attention. In particular, membranes have shown increased demand as a promising filtration technique for oily wastewater treatment. It has the advantages of being of low-cost and footprint, energy-efficient, and simple to use. However, the use of membranes in the filtration processes is limited by the development of foulants at the surface of the membranes, which negates the advantages of using membranes in oily wastewater separation [2]. Membrane fouling is developed by the buildup of oil, impurities, colloidal particles, bacteria, and others on the surface of the membranes or inside their pores, which impedes the penetration rate via the membranes, reducing, thereby, their productivity and raising the treatment costs [3,4]. Understanding membrane fouling during the filtration process increases our ability to control and mitigate its development. However, the complexity of produced water composition exacerbates fouling comprehension [5]. Membrane fouling can easily be detected via a concentration-based approach, direct observation of the membrane surface, gradient of TMP across the membrane [6], and critical flux methods [7]. To alleviate membrane fouling, it is first necessary to understand the interactions between oil droplets and the membrane to select the best combination of membrane units designed for oily wastewater treatment [8,9,10]. In addition, understanding the bulk feed stream droplet–droplet interactions at the surface of the membrane allows the selection of the best surfactant to stabilize the emulsion and lessen the development of membrane fouling [11,12]. 

Various methods have been explored to improve the membrane antifouling properties, enhance permeation flux, and increase the mechanical strength of membranes [13,14,15]. These include pre-treatment of the feed via dissolved air flotation [16], modification of feed characteristics (e.g., temperature, pH, ionic strength, salinity/conductivity), operating conditions optimization [17,18], hydraulic flushing [19], crossflow filtration [20], electrical, mechanical methods (e.g., vibration, ultrasonic, electric/magnetic [21,22,23,24]), and hybrid methods [25]. Most of these methods target reversible fouling. However, irreversible fouling is still a challenging drawback of membrane filtration. Physical antifouling methods also have been extensively explored to mitigate fouling and optimize the membrane performance. The most commonly used physical antifouling techniques are essentially hydrodynamic in nature, which comprise forward and reverse washing, backflushing, back-pulsing, and surface shearing [26]. Backflushing has become one of the standard procedures integrated with any crossflow filtration system for oily wastewater separation [27]. It can also be coupled with a pressure relaxation process or intermittent filtration to combat fouling as in activated sludge membrane bioreactors (MBRs) [28,29,30]. In this process, water reclamation is achieved by the settling of the sludge [25,26,27]. The pressure relaxation process is triggered when the system’s TMP reaches a maximum value. The TMP is released daily to target the concentration polarization (CP) without a concrete physical cleaning; then, a backwashing process is activated to clean the membrane by the permeate [31].

To be successful, we believe that an antifouling intervention must be administered before the membrane gets sufficiently covered with pinned droplets. Hydraulic cleaning is no longer efficient after irreversible fouling [30]. Consequently, the best way to reduce fouling is to dislodge anchored oil droplets by reducing their residency duration at the membrane pores before they build up the cake layer [30,31]. Furthermore, despite the enhanced knowledge in the field of membrane fouling control, recent research has emphasized on R&D of innovative strategies capable of providing sustainable membrane fouling alleviation. The current empirical study reports the use of the newly developed PFPT for the filtration of oily water systems using ceramic membranes. The PFPT novel physical antifouling method targets the roots of both reversible and irreversible fouling. This technique comprises a coupled filtration and cleaning in the same process. The periodic hydrodynamic perturbation is proposed by adjusting the pressure of the feed cyclically following a synchronous pulse train pressure form. Water permeation transports oil droplets to the membrane’s surface, but once the pressure is zeroed, the crossflow field releases pinned droplets. In this work, we test different TMP cycle time configurations to select an efficient pressure pattern that combats the fouling and enhances the overall membrane permeation. Finally, to establish the efficacy of this technique, a resistance model and membrane surface visualization were carried out to validate the approach’s antifouling properties and higher permeation flux for the ceramic membrane. While it was relatively easy to establish a modeling approach for the filtration of liquid emulsions using polymeric-type membranes based on the multicontinuum approach [32,33], there are specific challenges to extending this approach to the filtration of emulsions using ceramic membranes. 

## 2. Periodic Feed Pressure Technique

The idea behind the PFPT is to mitigate fouling by preventing the accumulation of oil droplets at the membrane surface during filtration. The fate of an oil droplet during filtration has been comprehensively studied using computational fluid dynamics (CFD) [34,35,36,37,38,39,40,41] and direct observation over the membrane surface [42] to understand the physics behind the oil settling and clogging of membrane pores. The results showed that an oil droplet, during the filtration process, undergoes one of the following four fates, namely (1) rejection, (2) permeation, (3) pinning, or (4) break-up. In each of these scenarios, four hydrodynamic forces have been identified as governing the fate of the oil droplet at the membrane pores. Figure 1 shows one such scenario of a droplet stressed by the crossflow field while permeating. 

The crossflow velocity exerts a hydrodynamic drag force that can, if sufficient, dislodge and transport the droplet towards the membrane module exit. Other forces that may have a minimal effect compared to the drag forces include the lift and buoyancy forces. As depicted in Figure 1, the two forces, namely due to hydrodynamic drag and interfacial tension, generate opposing torques that may balance under some critical conditions. At this moment, a critical velocity may be calculated that marks the onset of the breakup of a permeating droplet. That is when the torques generated by the drag and interfacial tension forces balance; this defines the critical crossflow velocity (CFV) beyond which a permeating droplet undergoes breakup; otherwise, the droplet continues permeation. The PFPT approach concept represents the cyclic influence of the applied permeation drag force. The TMP fluctuations (Figure 2) facilitate the release of the droplets pinned to the membrane’s surface by the action of the interfacial tension force. This TMP permutation causes detachment and dislodgement of the anchored oil droplets from the membrane pores, lowering the population of oil droplets at the surface of the membrane and resulting in a lessening of membrane fouling. When one of these forces is reduced or eliminated, the pinned oil droplets at the membrane pores are released and carried away by the crossflow mainstream field. PFPT aims simultaneously to combat membrane fouling and maintain a higher permeate flux. As a practical proof of concept concerning ceramic membranes, an experimental study is carefully designed for a crossflow ceramic membrane filtration system. The result has been analyzed and compared to the normal filtration process. In addition, the magnitude of fouling and fouling mitigation has been assessed using a resistance model to investigate the deposition and adsorption of oil droplets at the membrane surface for each experiment. A ceramic membrane visualization at the end of the operation time was performed to quantify the degree of the membrane fouling with/without the PFPT approach. Finally, the permeate water was collected and measured to compare the overall permeation flux for all processes.

## 3. Experimental Setup and Design

As stated earlier, the objective of this study is to bestow proof that PFPT [43] combats the fouling of ceramic membrane while maintaining a higher permeate flux than the normal filtration mode. We demonstrate that, contrary to standard filtration, the disturbances in permeation flux due to the cyclic change of the TMP expose oil droplets to unbalanced hydrodynamic forces that reduce the adherence of the droplets to the membrane surface. This is demonstrated in this study via extensive experimental works. 

Before the experimental tests began, the feed and membrane characterization were carried out. The feed characterization is determined by measuring the feed mean droplet size, zeta potential, chemical oxygen demand (COD), turbidity (TNU), pH, density, and viscosity. The membrane pore size, geometry and dimensions, thermal/chemical resistance, and permeability were defined. The TMP and CFV are the main operational parameters considered for the tests, whereas all other parameters are maintained constant. The coming section emphasizes the experimental design, measurement instruments, and precision.

### 3.1. Materials 

The Bakken oil from South Saskatchewan, Canada, with a density of 0.8872 g/cc and a viscosity of 5.23 cP, was used. Sodium dodecyl sulfate (SDS, 99 wt% pure) was purchased from Sigma-Aldrich (Saint Louis, MO, USA) and used as received for the feed synthesis. Sodium hydroxide (NaOH, >95 wt% pure) was obtained from EMD (Darmstadt, Germany), and phosphoric acid with a concentration of 85% was received from BDH Chemicals (Dubai, UAE) for the in-place cleaning of the ceramic membrane and the filtration unit after each experiment. Hydrochloric acid (HCl, SA431-500, 2N) was purchased from Fisher Chemicals (Hampton, NH, USA). Horiba (Kioto, Japan) S-316 # 100690 Extraction Solvent Oil (No. #5200100690, 75% polychlorotrifluoroethylene, 25% Chlorotrifluoro-ethylene Trimer) was bought and used as received. Ultrapure deionized water (DI < 5 ppb TOC and <0.1 colony-forming units of a microorganism/mL) was prepared from reverse osmosis (RO) water filtered by ultraviolet (UV) water purification system (EMD Millipore, 2012, Burlington, MA, USA). The 7-channel ceramic membrane (25 mm in diameter) was purchased from Tami industries, Nyons, France, and cut into pieces of 305 mm each. 

### 3.2. Feed Synthesis and Characterization

In this work, synthetic oily wastewater was prepared and used immediately for all the experiments to maintain the feed characteristics and to complete all the studies under similar conditions (Appendix A). It is usual in these types of experiments to utilize synthetic feeds due to their ease of preparation, availability, homogeneity, and constant properties. Two feed concentrations of 100 and 200 ppm were prepared to study the periodic feed pressure technique from the light Bakken oil with a viscosity of 5.23 cp (±1.0% accuracy) measured by Brookfield viscometer DV-II + Pro at 22.5 °C and a density of 0. 8872 g/cc (±5 × 10^−6^ g/cm^3^ accuracy, Kruibeke, Belgium) measured by an Anton Paar DSA 5000M digital densitometer (Montreal, Canada). A volume of 3.5 mL and 4.5 mL of light Bakken oil was added to 2 L of reverse osmosis water in the presence of 0.1 and 0.3 mM SDS as a surfactant to synthesize two feeds of 100 and 200 ppm, respectively. The oil content of the feeds was measured using HORIBA Oil Content Analyzer model OCMA-350 (±4 mg/L accuracy, Ontario, Canada). For each experiment, a volume of 24 L (12 batches) of synthetic PW was prepared. A single 2 L batch was also prepared by mixing oil, water, and surfactant for 2 min at 19,000 rpm (level 9.5 with variable pulses) using the commercial blender MX 1000 series (purchased from Waring Commercial, Stamford, CT, USA) to ensure higher-turbulence mixing and stability of oily wastewater emulsion. The pH of the feeds was also measured using a Horiba F-55 benchtop pH meter (Horiba 2003, with ±0.001 accuracy, Ontario, Canada) after its calibration with three pH buffer solutions points 4, 7, and 10, giving values of 6.331 and 5.945, respectively. 

The turbidity of the oily water emulsions was measured using a Hanna turbidity meter (model HI 83414, Hanna 2007, ±2% accuracy, Leighton Buzzard, England), giving 1431 and 1562 NTU, respectively. The chemical oxygen demand of the two feeds (concentrations of 100 and 200 ppm, respectively) to measure the oxygen required for the decomposition of organic matter and to oxidize inorganic chemicals was measured using a DRB200 Reactor and DR5000 UV-V spectrophotometer (London, Canada). Two COD Digestion reagent vials (HR 20, 500 mg/L) were used, one for a blank and the second for a sample. The blank and sample preparations were performed by adding 2 mL of deionized water and 2 mL of feed sample to each vial. The reactor preheated the two vials to 150 °C. The COD measurements for the 100 and 200 ppm feeds were 97 and 185 mg/L, respectively. Zeta potential and mean oil droplet size were measured using a Zetasizer Nano ZS (ZEN3600, Malvern 2009, Great Malvern, UK) for both feeds (100 and 200 ppm), giving values of −27, −33 mV, and 5.44, 4.25 μm, respectively. All the measurements were performed at room temperature at 25 °C. The Nano-ZS Zetasizer (Malvern 2009) measurement for zeta potential was performed using a disposable folded capillary cell (DTS1060C, ±0.6 mV accuracy, Great Malvern, UK); and for oil droplet size, a square polystyrene disposable cuvette was used (DTS0012, ±0.1 μm accuracy, Great Malvern, UK). During the measurement using the Zetasizer, the same refractive index (RI) and absorption index, and viscosity values of 1.45, 0, and 5.23 cP were added to measure the potential and the oil droplet size of the two emulsions, respectively. The refractive index of the prepared oily wastewater emulsions was measured at 25 °C using a model RX-5000 refractometer (ATAGO, Toronto, Canada). 

### 3.3. Ceramic Membrane Characterization 

The selection of the ceramic membrane has been made based on the membrane morphological properties, pore size, porosity, hydrophilicity, chemical/thermal resistance, and synthesized feed characteristics [44]. This work uses an ultrafiltration ceramic membrane with multi-channels, titania support, and an active zirconia layer for oily wastewater filtration. Figure 3 displays the ceramic membrane and its cross-sectional area. 

In Appendix A, Appendix B illustrates the membrane’s porosity measurement. Appendix A shows the longitudinal cross-section, and Table 1 lists the essential features and specifications of the considered ceramic membrane. The deposition process of the oil droplet at the ceramic membrane surface has been illustrated in Appendix A). The contact angle between the ceramic membrane and water and ceramic membrane and oil droplets in air demonstrates that the membrane surface wettability represents higher hydrophilic and oleophobic properties with a contact angle of ~35 and ~135°, respectively (see Appendix A). 

### 3.4. Description of the Filtration Unit and Filtration Process Design

The LabBrain CFU022 ceramic membrane filtration unit used to perform the experiments was purchased from LiqTech International, Hobro, Denmark (Figure 4). This filtration unit (Appendix A) can be operated as a semi dead-end or crossflow filtration system. The unit can operate manually or automatically in three modes: constant permeate flow, constant feed crossflow, and constant transmembrane filtration. The LabBrain unit is equipped with a membrane module lodging a ceramic element with dimensions of 25 ± 1 × 305 ± 1 mm. The lab unit is also equipped with a PLC (Siemens 6ES7 214- 1AE30-OXBO) that controls a loop valve (auto regulating valve) and two on/off valves from Bürkert, three solenoid valves from Festo, and a feed pump (Grundfos CRN 3–6) with a capacity of 5 m^3^/h at 2.5 bars. An air compressor (MotoMaster) with a tank capacity of 2.5 US gallons operated to a pressure of 6 bars supplies air to all units’ valves. In addition, the LabBrain contains one temperature transmitter to provide the concentrate temperature, and three pressure and flow transmitters to record each feed, permeate, and retentate, respectively. Before each experiment, the ceramic membrane was gradually soaked in deionized water and then drenched for an additional 12 h to completely displace the air from the membrane to achieve a high permeability water flux. The membrane element was installed in the housing, and feed (24 L) was prepared and added to the wastewater container to run the experiment.

All the crossflow experiments were performed in batch mode for 2 h. The permeate was collected to be weighted, and the retentate was completely returned to the feed tank. The reverse osmosis water with the same collected permeate volume was added continuously to the feed to maintain the same feed concentration. To set up the operating conditions at crossflow (CFV) of 1 m/s and transmembrane (TMP) of 1.5 bar, the speed pump was increased gradually, and the retentate valve was adjusted to the correct opening percentage through the touchpad. Once the pressure in the unit was stable, DATALOG was switched on, and all the data, including TMP, CFV, temperature, valves’ opening percentages, feed flow rate, permeate flow fate, and retentate flowrate, were automatically logged every 3 s. The permeate was collected in a beaker, and a permeate sample of 4 mL was used to analyze its oil content at the end of the experiment. 

### 3.5. Experimental PFPT Design 

To enhance the crossflow cleaning potential while the system was in operation, the PFPT approach was applied. The aim was to improve membrane cleaning by preventing the deposition, accumulation, clustering, and coalescence of the oil droplets at the membrane surface using a fluctuating conjunction mode of filtration and cleaning cycles. 

The PFPT cycle is referred to as PFPT (filtration time-cleaning time), where time is in minute units. The PFPT patterns employed in this study are explicitly 1-1 cycle, 1-2 cycle, and 2-2 cycle (Appendix A). The applied PFPT was set up by an on/off valve (03V02, Angle seat valve) from Bürkert that controls the permeate flow. The process was controlled and operated with a small integrated operating cabinet consisting of a touch screen. All the set/actual features and values were displayed and could be operated from the touchpad. To switch between an on/off filtration mode in PFPT, the pneumatic valve is turned on/off. In the off-filtration, the pressure drops gradually, as depicted in Appendix A), until the TMP reaches zero bar, where the membrane surface is cleaned, and oil droplets are displaced with the mainstream (Appendix A). At the same time, when the valve is switched on, the applied TMP immediately rises to the operating value of 1.5 bar (Appendix A), and the filtration mode is switched on again. For the verification and validation of the PFPT concept, the experiment was carefully designed to analyze the effect of permutation of the pressure patterns on the ceramic membrane performance, membrane rejection capacity, and fouling mitigation. Finally, the PFPT results were compared to the regular filtration process. Visual inspection of the ceramic membranes post-filtration and post-water washing was performed to highlight the fouling control during each experiment (Appendix A). The flux was continuously monitored under the same operating conditions during the experimental time of 2 h, and all experiments were performed in duplicate.

### 3.6. Membrane Oil Rejection

As previously discussed, the membrane performance was studied using the LabBrain crossflow filtration unit (LiqTech). The membranes were left to soak in DI water and then pre-pressurized within the reasonable limits required by the supplier. The 24 L of synthetic feed oily wastewater emulsion were prepared for each experiment for 2 h. The permeate water was collected during all the experiments. A sample of 4 mL of permeate water was taken at the end of the experiment to measure its oil content (Appendix A). The oil content of the oily wastewater synthetic feeds (Appendix A) and the permeates were measured using an environmentally safe IR spectroscopy grade S-316 extraction solvent and a HORIBA-model OCMA-350 oil content analyzer (Appendix A). Before measurements, the OCMA-350 oil content analyzer was calibrated through two point calibration. A zero-shift value (0.0 mg/L) was prepared using a zero liquid S-316 (Specific gravity of 1.75 g/mL, at 20 °C). The span calibration value (200 mg/L) was performed using a mixture of 2:1 *v*/*v* of B-heavy oil (Specific gravity 0.895 g/mL, at 20 °C) in a solvent S-316.

Primarily, the sample was prepared to pH below 2 by adding 1 mL of hydrochloric acid solution (2 N). Then, a double volume of the extraction solvent S-316 was added to the sample in the vial. The mixture was agitated vigorously for one minute and left for another minute to settle. Two-phase layers of oil/solvent (at the bottom) and water (at the top) were separated (Appendix A). The oil/solvent phase extracted and its oil content in the sample was measured. The oil rejection of the ceramic membrane was calculated using the following equation:(1)$$R=\left(1-\frac{C_{p}}{C_{f}}\right)\times \;100\%$$
where *C_p_* and *C_f_* are the concentrations of the permeate and oily water feed, respectively.

## 4. Results and Discussion

Experimental work has investigated how the applied PFPT affects the ceramic membrane filtration process for oily water systems. The experiment has been designed to run with two feed concentrations, namely 100 and 200 ppm, to depict the effect of the feed oil content on the PFPT. Initially, the ceramic membrane was conditioned, and a permeability test was performed. Second, the feed filtration under the same operating conditions (TMP = 1.5 bar and CFV = 1 m/s) was set to measure the performance of the ceramic membrane and its oil rejection capacity during an overall operation time of two hours. Third, the novel PFPT approach was applied for the two oily water emulsions under the same operational parameters (Appendix A). Fourth, a performance/efficiency comparison between the normal filtration (i.e., no PFPT) and the PFPT scenarios was carried out by measuring the overall permeate volume and comparing the permeate flux decline. Finally, a visual inspection of the ceramic membrane’s internal active wall surface and flow resistance analysis was accomplished to measure the magnitude of the fouling mitigation of the membrane element with/without PFPT.

In the normal filtration process, a quick flux decline of about 85% for both feeds was detected within the first 10 min of the filtration time [46,47]. The adsorption and accumulation of the oil droplets at the membrane surface resulted in a gradual growth of the overall hydraulic resistance. After one hour of filtration, a steady-state dynamic equilibrium was achieved between the membrane fouling and surface shearing crossflow along the membrane surface [48]. Figure 5 illustrates that the ceramic membrane normalized the decline in permeate flux for the two feeds (i.e., the 100 and the 200 ppm emulsions) during the normal filtration as a function of time.

Pictures in the Appendix A visually display the degree of the fouling development on the internal membrane channel surface at the end of the experiment for both the oily water emulsions. The results showed the limitation of the hydrodynamic crossflow field during the normal filtration mode to effectively combat and mitigate fouling development [43,44]. Appendix A also illustrates that the ample post-operation RO water cleaning could not return the membrane to its initial state by cleaning and detaching the deposited oil at the membrane surface.

Figure 6 and Figure 7 show the flux pattern during an applied 1-1 PFPT for both emulsions (i.e., the 100 and 200 ppm system). The 1-1 PFPT refers to a permeation and a cleaning half-cycle of 1 s each. The flux behavior during the 1-1 PFPT, as depicted in Figure 6a and Figure 7a, totally differs from that of a regular non-PFPT filtration. In each PFPT cycle period, the permeate flux starts from the highest normalized value of 1. It then declines to reach an approximately stable value before the cleaning mode is activated, where the permeate flux drops until it is nulled. It is essential to mention that through the cleaning mode, it was observed that the TMP slowly declined towards zero, as shown in the pressure profile (Appendix A).

Before the applied pressure reached 0 bar, the transmembrane pressure (TMP) remained higher and able to drive permeation. This explains the scattered data in Figure 6a and Figure 7a, which show a continuous permeation due to the positive pressure difference across the membrane. When the applied pressure becomes very low, approximatively null, permeation drag stops, and the crossflow drag governs. In this case, the complete cleaning process takes place. This is presented by the bottom scattered data readings that illustrate no permeation flux. One can generally categorize the data presented in Figure 6a and Figure 7a into three regions, namely, (1) the top region, which displays the permeation flux during the filtration mode, (2) the middle-scattered data region illustrates the permeation flux during the pressure decline period, and (3) the bottom region where a complete cleaning is in progress. The majority of the scattered permeation data is above 40% of pure water permeate flux during the entire experiment. This is in contrast to the behavior during the normal filtration mode, where the flux declines below 85% after just a few minutes of operation. Figure 6b and Figure 7b illustrate the state of the membrane just after the cleaning half-cycle during a 2-h experiment for the 100 and 200 ppm feeds, respectively. These figures show that after each PFPT cycle, the membrane returns to its original clean state. The normalized flux data manifest this at the start of each PFPT cycle which is essentially 1. It is to be noted, however, that the cleaning efficiency of the membrane is more pronounced when the concentration of the oil in the feed emulsion is low. In other words, for the 200 ppm emulsion, the 1-1 PFPT does not return the membrane to its original clean state during the two hour experiment, Figure 7b, as was the case for the 100 ppm emulsion, Figure 6b. This indicates some fouling, albeit far less than observed in non-PFPT normal filtration. It also implies that, for each feed emulsion, there may exist an optimum PFPT cycle.

Similarly to the 1-1 PFPT cycles, Figure 8a and Figure 9a show the profile of the normalized permeate flux during the two hour experiment for the 1-2 PFPT for the two feeds of 100 and 200 ppm, respectively. During the 1-2 PFPT, one notices that the density of the scattered flux data is higher towards the top region than that of the 1-1 PFPT, implying higher permeate flux. The higher filtration in the 1-2 PFPT can be explained by the doubled cleaning time compared to the 1-1 PFPT. It may, therefore, be possible to generally state that the longer the cleaning time, the cleaner the membrane surface. Furthermore, Figure 8b and Figure 9b show that the filtration starts after each half-cycle with the membrane almost clean throughout the two hours’ operation, particularly for the 100 ppm scenario. For the 200 ppm scenario, the permeation flux slightly drops, indicating some fouling development, albeit small.

Figure 10a and Figure 11a display the performance of the filtration system upon adapting the 2-2 PFPT. In this case, the PFPT is designed to double the permeation and the cleaning half-cycles compared to the 1-1 PFPT. The reason for running this case is to understand the impact of prolonging the permeation half-cycle on the membrane performance. In the 1-1 PFPT, the cleaning process did not last the whole one minute due to the relatively slow decline of the applied pressure when the filtration was turned off. However, during the 2-2 PFPT, the TMP reached its minimum value of zero bar, and the cleaning cycle took more than a minute. This confirms that the dragging crossflow will likely have sufficient time to dislodge unstable oil droplets compared to the 1-1 PFPT scenario. The continuous cleaning prevents the growth of a fouling resistance layer by destabilizing the oil droplets and preventing them from seeding the coming oil droplets when the filtration mode is activated. A clean membrane surface guaranteed less hydraulic resistance and higher permeate flux. However, to evaluate the state of the membrane for each complete cycle, it is most important to look at the flux at the start of each cycle. As shown in Figure 10b and Figure 11b, it is clear that the membrane starts primarily as a new membrane (normalized permeate flux equals 1). In other words, the cleaning half-cycle effectively detaches pinned droplets away from the membrane surface via the crossflow field. Even though there are instances in which the flux does not start from the state of a complete clean membrane, implying some degree of fouling, the membrane is never far from a clean start state.

In summary, Figure 6, Figure 7, Figure 8, Figure 9, Figure 10 and Figure 11 display the ceramic membrane flux decline pattern when applying the PFPT. The flux-decline behavior in PFPT is entirely different from normal filtration. Embedding a cleaning time during the filtration process improved fouling mitigation appreciably. The periodic feed pressure technique thwarts the oil droplet accumulation on the membrane surface when the cleaning is in operation; the hydrodynamic environment near to the oil droplet changes because the applied transmembrane pressure becomes nearly zero. Consequently, the pressure-driven process that causes permeation becomes a crossflow cleaning process due to the negligent effect of the permeation drag force that drives and maintains oil droplets at the membrane pores. In this instance, the crossflow drag detaches the unstable oil droplets to sweep them away by the crossflow watercourse.

For permeate comparisons, the average PFPT permeate flux profiles (averaged over the filtration half-cycle, i.e., when the TMP is active) were considered, and the half-cycle cleaning data were ignored. Figure 12 and Figure 13 illustrate the normal filtration (i.e., no PFPT) and the PFPT-normalized permeate flux data during the experimental time. PFPT positively impacted the membrane permeability and the overall oil rejection efficiency during the half-cycle filtration. Appendix A) represent the oil content of the permeate for the regular filtration and each PFPT scenario for both feeds. PFPT achieved an oil retention efficiency of >97% compared with the normal filtration process, which reached >91% for both feeds. Appendix A) provide the turbidity measurements for the feeds and permeate in each process.

Figure 12 illustrates the normalized permeate flux behavior for the three PFPT cycles when the oil content of the feed was 100 ppm. The 1-2 PFPT describes the best permeation flux, followed by the 2-2 PFPT, and lastly, the 1-1 PFPT. This implies that the shorter the filtration half-cycle and the longer the cleaning half-cycle, the better the performance. The 1-2 PFPT showed a ceramic membrane permeability of about 80% of the clean water permeation flux at the end of the experiment. This is a manifestation of the cleaning half-cycle that is twice that of filtration time, which maintains the membrane surface as new.

Similarly, the 2-2 PFPT and the 1-1 PFPT achieve permeability recovery above 76% and 65%, respectively. This is in contrast to the no PFPT scenario in which the permeation flux declines to below 85% of the permeation flux at the start. The limited effect of the crossflow field to control the fouling and the continuous oil droplet deposition on the membrane surface form a growing fouling resistance layer which is evident by the significant decline in permeate flux. Figure 13 illustrates similar behavior for the 200 ppm feed. The permeate flux recovery for the 1-2, 2-2, and 1-1 PFPT are 80%, 62%, and 57% of the pure water permeate flux, respectively. The overall behavior is that the normalized permeate flux for the PFPT is better than that of normal filtration for ceramic membranes.

In summary, PFPT aims to minimize membrane fouling while retaining a greater permeation flux. Figure 12 and Figure 13 illustrate that the permeate flux for all PFPT cycles is higher than the normal filtration, despite the PFPT flux curves presenting only 50% of the filtration time in both the 1-1 and the 2-2 PFPT and 33% in the 1-2 PFPT. In addition, Figure 12 and Figure 13 show that the 2-2 PFPT has better normalized permeate flux recovery than the 1-1 PFPT by almost 10%. This may be due to the applied pressure cycle in the longer cleaning time for 2-2 PFPT’s.

It is interesting at this time to pose the following questions: might less filtration time in PFPT cycles affect the overall permeate amount? does the PFPT combat the fouling and affect the membrane’s overall productivity? To answer these questions, the total amount of the permeate was measured during each experiment, and the overall permeate was compared to select the best filtration process under the operating conditions of TMP of 1.5 bar and CFV of 1 m/s. For this purpose, ceramic membrane performance during the PFPT was measured in terms of the permeate volume. As shown in Figure 14, the results indicate that the 1-2 PFPT produced the largest permeate volume at the end of the experiment compared to the 2-2 PFPT, the 1-1 PFPT, and the No PFPT.

In conclusion, the overall filtration and cleaning time tremendously affect membrane productivity. When the cleaning half-cycle was longer than the filtration half-cycle, the ceramic membrane remained clean. Despite the shorter filtration time in PFPT due to the cyclic embedded cleaning mechanism, PFPT is rewarded with less fouling resistance and a high permeation flux compared to the regular filtration mode. To demystify the membrane antifouling behavior during PFPT cycles, visual inspection of the membrane fouling and resistance analysis is extensively studied in the following section.

## 5. Resistance Profile Analysis

The membrane fouling development and mitigation during the normal filtration and PFPT scenarios have been studied using fouling resistance distributions. To picture the progressive fouling in a crossflow ceramic membrane filtration, the resistances were computed using experimental data and used as an operational parameter to evaluate the membrane fouling. The permeate flux decline was associated with various resistances-in-series, which are explicitly identified in three primary resistances: membrane resistance (*R_m_*), cake layer resistance (*R_c_*), and internal intrinsic resistance (*R_i_*) [49]. Figure 15 and Figure 16 are the experimental resistances measured according to the membrane cleaning procedure (Appendix B, Figure A1 and Figure A2). Our study was based on the assumption [50,51,52,53] that the cake resistance (*R_c_*) was removed by circulating the RO water through the membrane, and internal resistance (*R_i_*)—caused by the internal trapped oil within the ceramic membrane pores—was cleaned by alkaline/acid cleaning (Appendix B, Table A1). Membrane ceramic resistance (*R_m_*) is calculated based on the permeability test of the new membrane (Appendix B, Figure A3), and the total fouling resistance (*R_t_*) was computed using Darcy’s law (i.e., $J=TMP/R_{t}$). The total fouling resistance can also be calculated as the summation of all the resistances (i.e., $R_{t}=R_{m}+R_{c}+R_{i}$). Figure 15 and Figure 16 show that during the crossflow ceramic membrane filtration, the dominant fouling resistance is the cake layer resistance. The internal and membrane resistance had a lower effect on the overall fouling resistance [54,55,56]. Therefore, the cake resistance is considered an operational parameter to evaluate the extent of the membrane fouling [57,58]. For the PFPT, the cake resistance layer is visibly smaller than membrane resistance, explaining why the fouling is completely mitigated. Thus, the novel PFPT created a hydrodynamic environment at the membrane surface that limited the fouling development.

Consequently, PFPT supports controlling the membrane fouling without a potential filtration system modification or a rise in energy consumption [59]. PFPT is a novel antifouling technique that can be implemented in any filtration unit to combat and mitigate membrane fouling without additional cost. Overall, PFPT (1-2) cycle was considered the best combination of filtration/cleaning setup to control the ceramic membrane fouling for oily wastewater treatment under our operating conditions.

## 6. Conclusions

The newly developed novel and environmentally friendly periodic feed pressure antifouling technique (PFPT) has been used in this study to combat the fouling of ceramic membranes. This technique is based on generating a cyclic transmembrane pressure to prevent aggregation and adsorption of oil droplets at the membrane surface. The PFPT is designed by alternating the transmembrane pressure (TMP) between zero and the operational pressure to detach pinned oil droplets and transport them by the crossflow field. The applied pressure fluctuation minimizes the permeation drag’s effect while maintaining the crossflow drag. It becomes easier, therefore, for the crossflow drag to dislodge and detach pinned and permeating droplets. This considerably mitigates and controls the fouling by shortening the oil droplets’ deposition time at the membrane surface and preventing them from seeding, clustering, and coalescing with the incoming ones when the pressure is switched to its highest value.

Three feed pressure cycles have been tested in a filtration-to-cleaning time ratio of 1:1, 1:2, and 2:2 and the results compared to the standard continuous filtration process. The finding proves that despite the filtration time being shortened by adding a cleaning time, the permeate flux recovery was very high, and even better than in regular filtration mode. The visual inspection of the ceramic membrane post-filtration depicted that all the PFPT cycles kept the ceramic membranes clean as new after the operation time of 120 min. The overall permeate volume for each filtration with/without PFPT was collected and compared to highlight the filtration performance of a ceramic membrane undergoing the PFPT. It is essential to mention that during the PFPT, the initial permeation flux was recovered under the same operating conditions and without any modification of membrane characteristics. Furthermore, the investigation of the fouling development was studied using a resistance model accompanied by a post-operation visual inspection of the ceramic membrane surface. The results showed that the cake layer resistance was dominant in normal filtration mode compared to all PFPT cycles. Conversely, the PFPT process displayed lower resistance profiles (reversible/irreversible resistances) and clean membrane surfaces compared to those experienced in normal filtration mode.

Energy costs are calculated based on the energy required to pump the feed through the membrane system (which is the energy needed for filtration). In PFPT, the filtration time is reduced to 50% in PFPT 1-1 and 2-2 and 33% in PFPT 1-2 compared to the total time of 2 h of filtration in no-PFPT. This can explain why the energy consumed through normal filtration is higher than that consumed in PFPT mode. In addition, when the fouling occurs, the resistance increases, which requires an increase in transmembrane pressure to overcome the overall membrane resistance to ensure continuous filtration. While in PFPT mode, the filtration time is shortened, and the crossflow field is used periodically for cleaning when the transmembrane pressure declines.

In conclusion, the PFPT innovative crossflow filtration technique is an environmentally friendly method characterized by an advanced reversible/irreversible fouling mitigation, higher permeation flux capacity, easy process implementation, energy consumption saving, and no experimental filtration setup upgrade or modification.

## Figures and Tables

**Figure 1 membranes-12-00868-f001:**
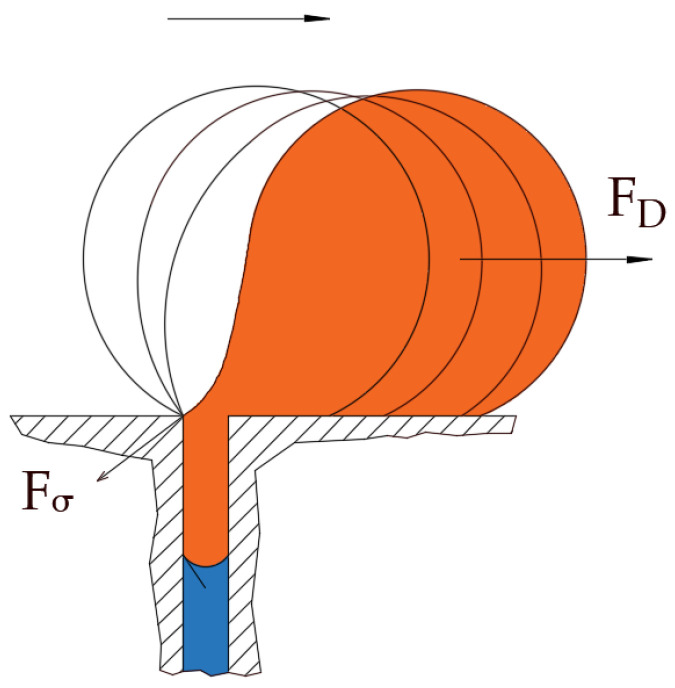
Schematic of the hydrodynamic and interfacial tension forces acting on a pinned oil droplet undergoing permeation in crossflow filtration. F_D_ is the drag force due to the crossflow field, and F_σ_ is the interfacial tension force.

**Figure 2 membranes-12-00868-f002:**
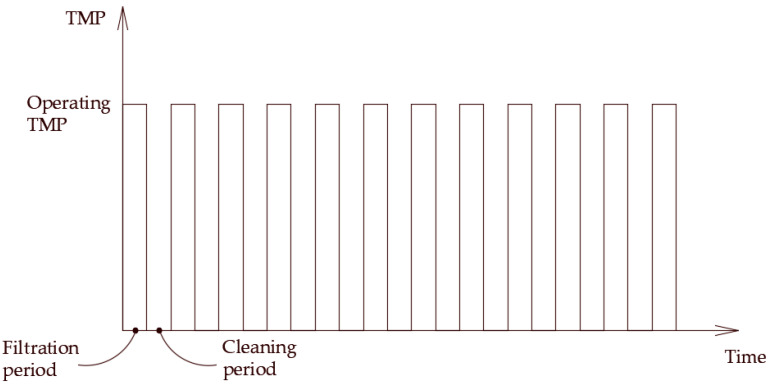
Schematic representation of the periodic feed pressure.

**Figure 3 membranes-12-00868-f003:**
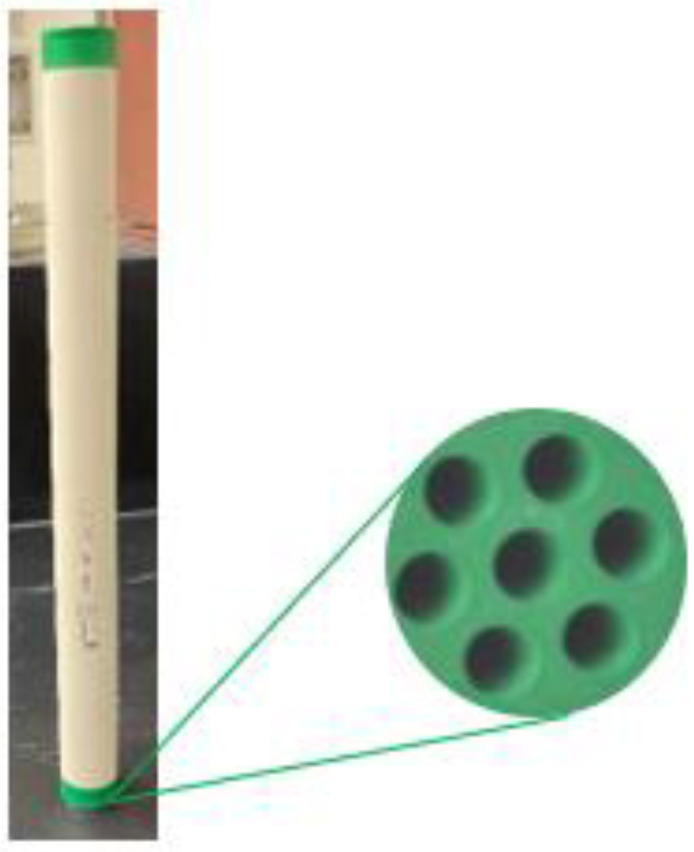
Ceramic membrane representation and its cross-sectional area.

**Figure 4 membranes-12-00868-f004:**
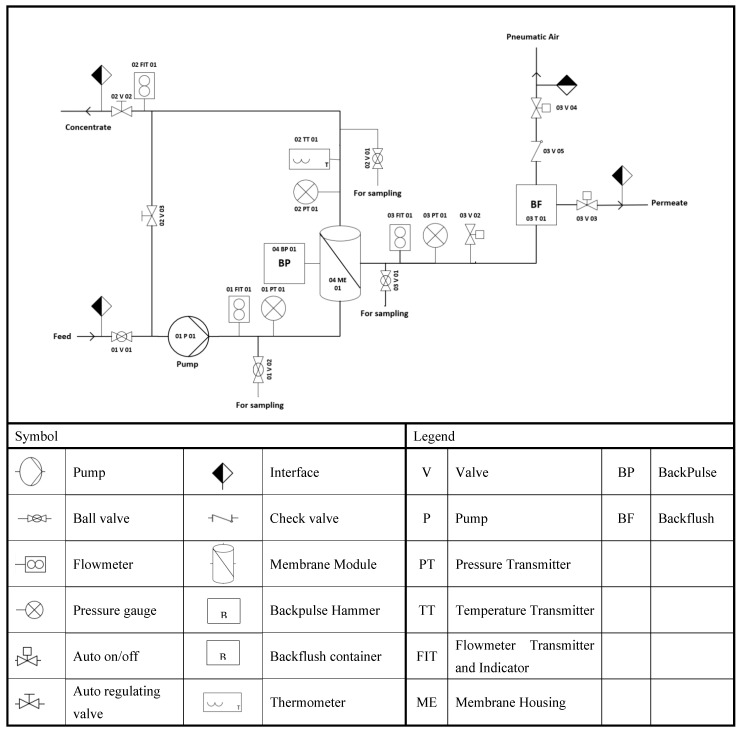
LabBrain P&I Diagram, adapted from [45].

**Figure 5 membranes-12-00868-f005:**
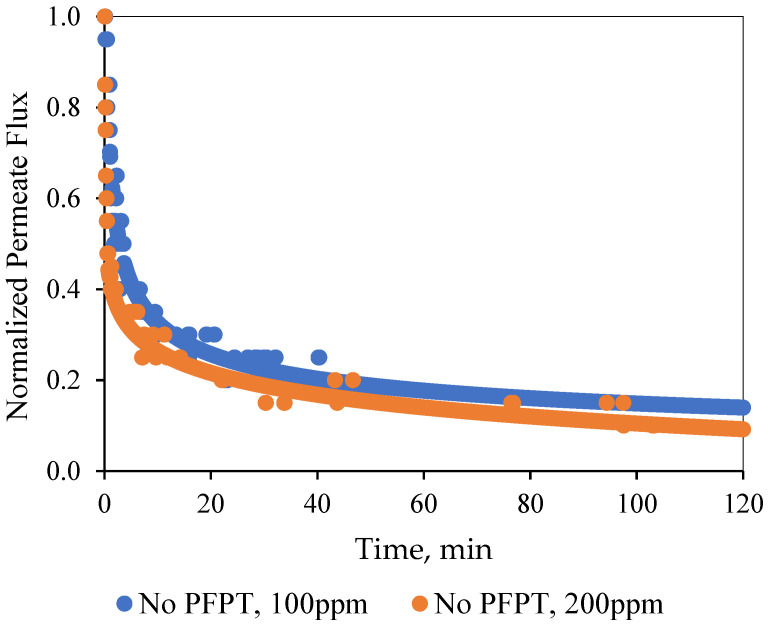
Normalized permeate flux behavior as a function of time for normal filtration mode at TMP: 1.5 bar and CFV: 1 m/s.

**Figure 6 membranes-12-00868-f006:**
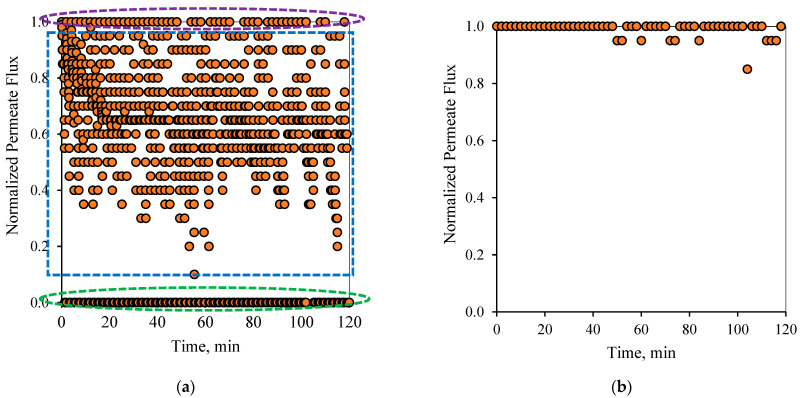
Permeate flux profile using the 1-1 PFPT for a feed of 100 ppm at CFV: 1 m/s: (**a**) PFPT filtration mode, (**b**) membrane permeability after each cleaning half-cycle. (1) region 1 (purple), (2) region 2 (blue), (3) region 3 (green).

**Figure 7 membranes-12-00868-f007:**
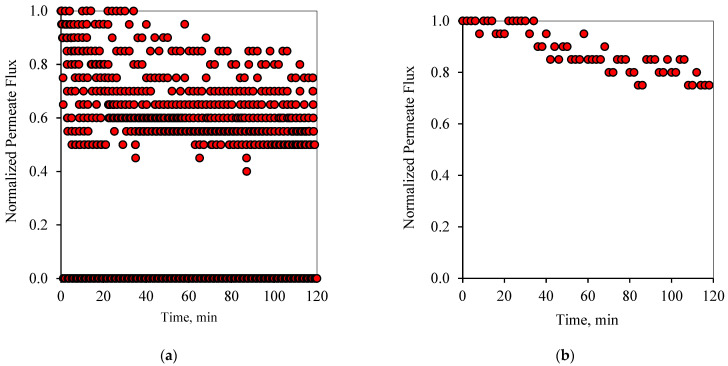
Permeate flux profile using the 1-1 PFPT for a feed of 200 ppm at CFV: 1 m/s: (**a**) PFPT filtration mode, (**b**) membrane permeability after each cleaning half-cycle.

**Figure 8 membranes-12-00868-f008:**
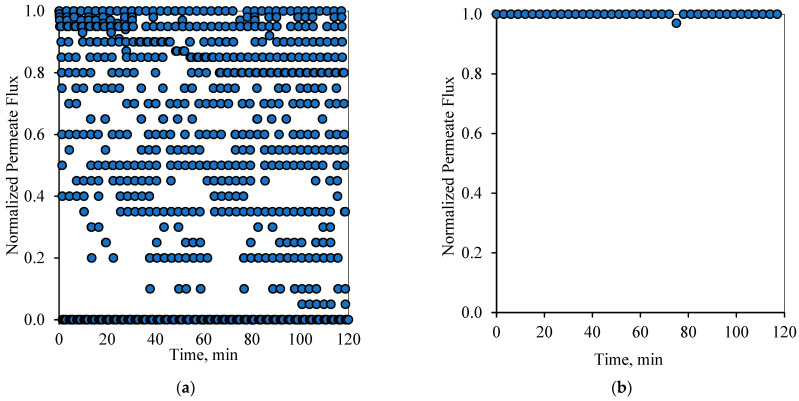
Permeate flux profile using the 1-2 PFPT for a feed of 100 ppm at CFV: 1 m/s: (**a**) 2 h of filtration operation, (**b**) membrane permeability after each cleaning half-cycle.

**Figure 9 membranes-12-00868-f009:**
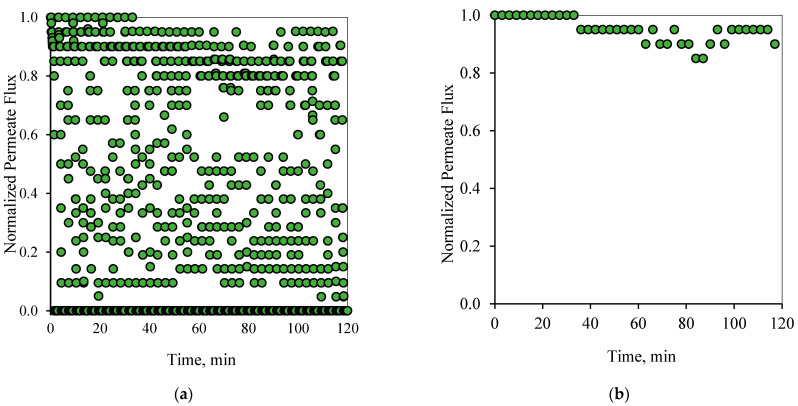
Permeate flux profile using the 1-2 PFPT for a feed of 200 ppm at CFV: 1 m/s: (**a**) 2 h of filtration operation, (**b**) membrane permeability after each cleaning half-cycle.

**Figure 10 membranes-12-00868-f010:**
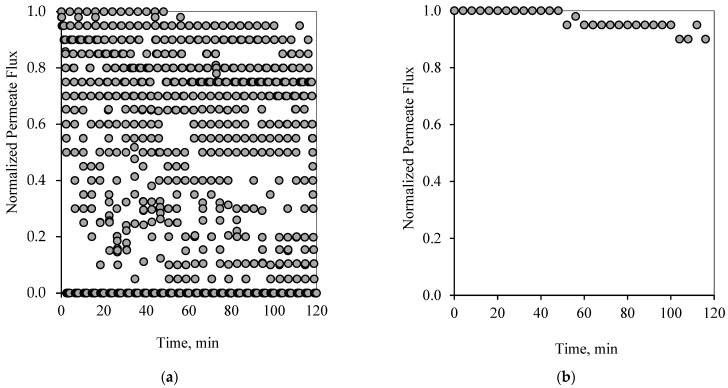
Permeate flux profile using the 2-2 PFPT for a feed of 100 ppm at CFV: 1 m/s: (**a**) 2 h of filtration operation, (**b**) membrane permeability after each cleaning half-cycle.

**Figure 11 membranes-12-00868-f011:**
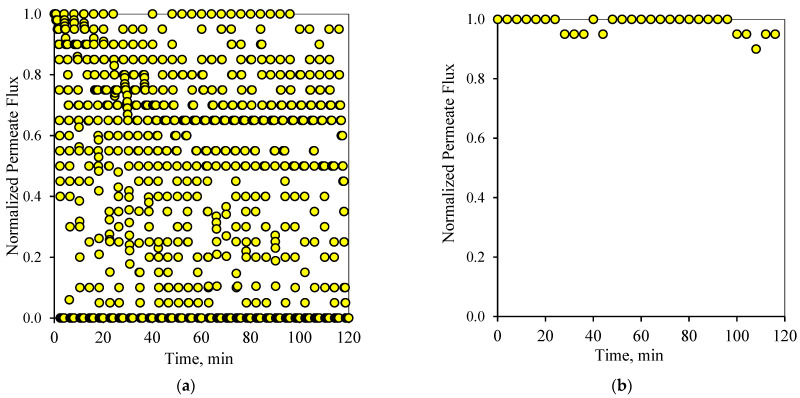
Permeate flux profile using the 2-2 PFPT for a feed of 200 ppm at CFV: 1 m/s: (**a**) 2 h of filtration operation, (**b**) membrane permeability after each cleaning half-cycle.

**Figure 12 membranes-12-00868-f012:**
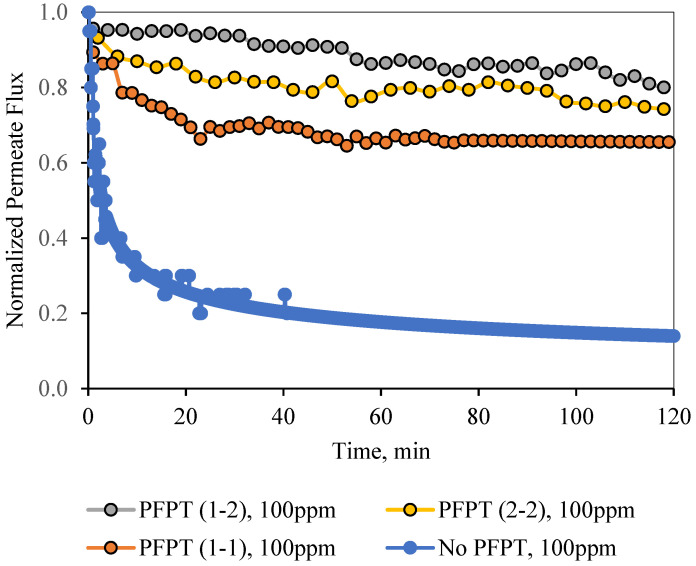
Normalized permeate flux decline for normal filtration and the 1-1, 1-2, and 2-2 PFPT as a function of time for a feed of 100 ppm.

**Figure 13 membranes-12-00868-f013:**
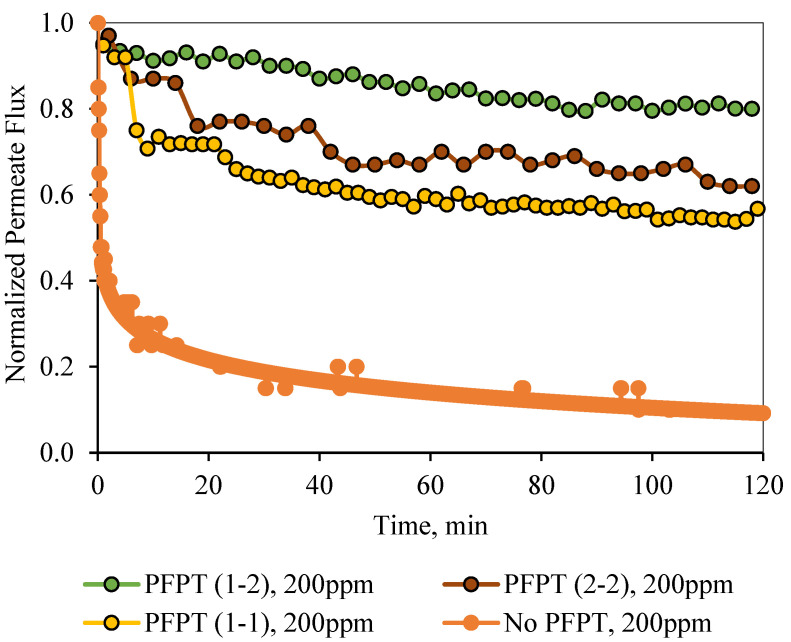
Normalized permeate flux decline for normal filtration and the 1-1, 1-2, and 2-2 PFPT as a function of time for a feed of 200 ppm.

**Figure 14 membranes-12-00868-f014:**
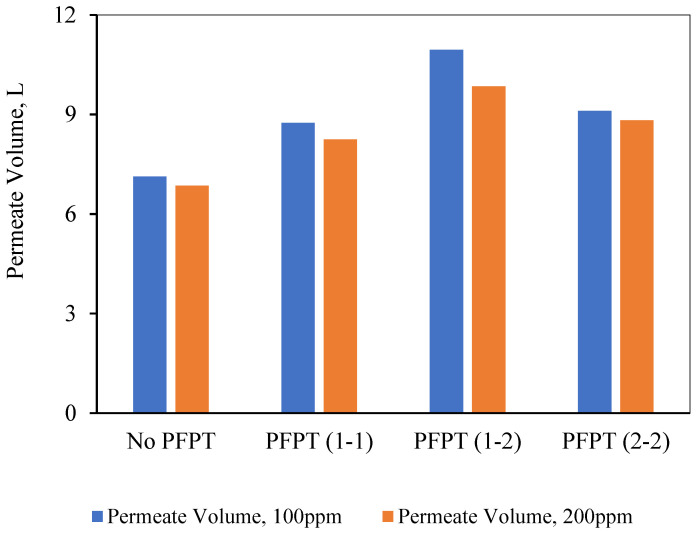
Comparison of the overall permeate volume for the No PFPT and the three PFPT cycles after a 2-h experiment.

**Figure 15 membranes-12-00868-f015:**
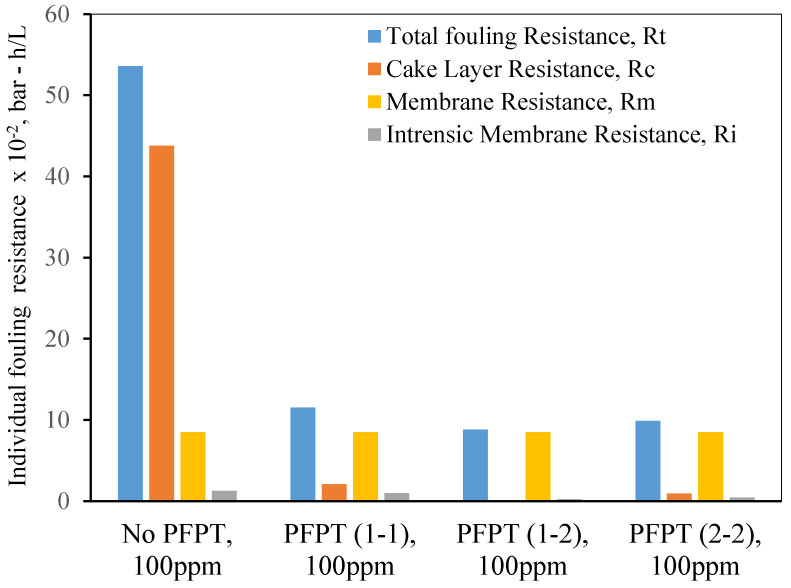
Individual fouling resistances for standard filtration and PFPT for a feed of 100 ppm at the TMP: 1.5 bar and CFV: 1 m/s.

**Figure 16 membranes-12-00868-f016:**
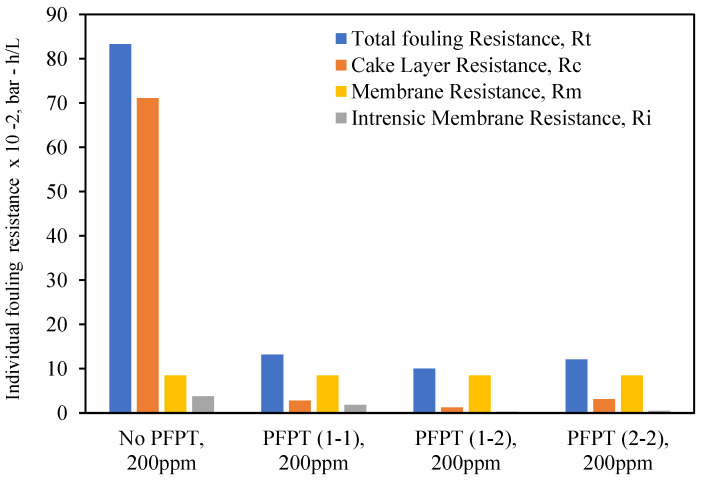
Individual fouling resistances for standard filtration and PFPT for a feed of 200 ppm at the TMP: 1.5 bar and CFV: 1 m/s.

**Table 1 membranes-12-00868-t001:** Ultrafiltration ceramic membrane characteristics.

Membrane	Characteristics
Materials	Support: Multi-channel Titanium Oxide (TiO_2_) Active layer: Zirconium Oxide (ZrO_2_)
Bursting pressure	>90 bar
Maximum working pressure	10 bar
Best operating pressure	3 bar
pH range	0–14
Max operating temperature	<250 °C
Thermal shock resistance	ΔT° instantaneous <60 °C *
Steam sterilization	121 °C-30 min
Pore size/MWCO	150 kg/mol
Dimensions, mm	25 ± 1 × 305 ± 1
Number of channels	7
Hydraulic diameter of channels, mm	6 ± 0.1
Filtration area, m^2^	≈0.04186 ± 0.006
Cross-sectional area, m^2^	0.001172 ± 0.006
Membrane regeneration (base)	NaOH, 5 g/L (85 °C, 30 min)
Membrane regeneration (acid)	HNO_3_, 5 mL/L (50 °C, 15 min)

* Temperature difference between liquid and membranes.

## Data Availability

Data or models used during this study are obtainable from the corresponding author by request.

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
