# Peer review of "Experimental Investigation of the Novel Periodic Feed Pressure Technique in Minimizing Fouling during the Filtration of Oily Water Systems Using Ceramic Membranes"

_membranes, 2022, doi:10.3390/membranes12090868_

Round 1

Reviewer 1 Report

The manuscript experimentally investigated the fouling mitigation of oil/water system using ceramic membranes using periodic feed pressure technique (PFPT). The following points should be addressed before the manuscript is accepted for publication

1.       Experimental results should be coupled with a relevant analytical model

2.       In the last part of the introduction the novelty and significance of the study must be presented

3.       It should be presented that why ceramic membranes were preferred over other membranes.

4.       The following latest literature for oil water should be included:

a.       https://doi.org/10.1016/j.jwpe.2021.102293

b.       https://doi.org/10.1016/j.memsci.2011.03.011

c.       https://doi.org/10.1016/j.seppur.2021.118581

d.       https://doi.org/10.1021/acs.langmuir.1c02046

e.       DOI: 10.1016/j.cherd.2018.12.007

f.        https://doi.org/10.1016/j.memsci.2013.06.053

g.       https://doi.org/10.1016/j.apsusc.2018.04.180

h.       DOI: 10.1016/j.desal.2020.114428

Reviewer 2 Report

One of the important question comes to my mind is that whether this process periodic feed pressure technique (PFPT)can be applied for cleaning and reuse of the membranes.  The other query comes to my mind is that what about the energy involved for this cycling filtration and PFPT process?.

If there is any references related to the above queries and can be addressed in the revised manuscript, it would be useful to the researchers, strengthen the manuscript and also citation of the manuscript.

The below mentioned sections must be filled in the revised manuscript.

Supplementary Materials:  Author Contributions: Funding:  Data Availability Statement:  Conflicts of Interest

After the above corrections, the manuscript can be accepted for publication.
